# The Predictive Factors of New Technology Adoption, Workers’ Well-Being and Absenteeism: The Case of a Public Maritime Company in Venice

**DOI:** 10.3390/ijerph182312358

**Published:** 2021-11-24

**Authors:** Chiara Panari, Giorgio Lorenzi, Marco Giovanni Mariani

**Affiliations:** 1Department of Economics and Management, University of Parma, 43126 Parma, Italy; 2AVM Holding Venice, 30135 Venice, Italy; giorgio.lorenzi@actv.it; 3Department of Psychology, University of Bologna, 40100 Bologna, Italy; marcogiovanni.mariani@unibo.it

**Keywords:** acceptance of technology, job satisfaction, work engagement, absenteeism, shipping

## Abstract

The main goal of this research was to investigate the psychosocial aspects that influence the acceptance of innovative technology in maritime transport and its impact on employees’ work-related wellbeing and absenteeism. In particular, this study focused on a device that had been introduced to sailors working in water public transportation in Venice. The theoretical framework included two integrated models: the TAM model, concerning acceptance of the technology, and the JD-R model, related to workers’ well-being. A two-wave study was conducted; at T1, a self-report questionnaire was administered to 122 sailors. Four months after its first administration (T2), objective data related to days of absenteeism were collected. The study showed that the perceived ease of use and the usefulness of the device influenced the workers’ intentions to use the technology and their motivational processes of work engagement, which was also related to social support. Work engagement impacted on work satisfaction and predicted the level of absenteeism (measured at Time 2). The implementation of a new technology may fail if transportation companies do not consider psychosocial factors that assist in the acceptance of such technology and promote the involvement of workers in the technological system.

## 1. Introduction

Technology plays a vital role in organizations, as it increases performances and productivity [1], and is crucial for rapid organizational responses to the outside environment. This is true for virtually all enterprises on which technology has a strong impact, not only on performance, but also on safety—including sea transport [2], which is the context of this study. The maritime industry is one of the most advanced users of information and communication technology. Navigators make use of technology in the form of maps, global position systems (GPSs), detectors and a variety of other functions, in order to plan travel routes and find destinations. These systems also increase safety and allow navigators to improve energy efficiency [2]. The introduction of new technology in maritime organizations not only enables/leads to the provision of better quality services to customers, but can also ensure efficient processes with higher safety levels, considering the strong impact of accidents involving ships [3].

Before the crisis linked to the COVID-19 pandemic, the shipping industry contributed to economic development by transporting 90% of the global cargo volume across the world [4], as well as 415 million passengers in Europe alone [5]. In 2021 the situation, due to the impact of COVID-19 on shipping traffic, is still evolving. The number of stopovers between 2019 and 2020 was −52%, and between 2019 and 2021 it was −14% [6]. For this reason, in the post-Covid era, technology may play an even more crucial role in rethinking the way these organizations function. Nowadays, transport and logistics networks are becoming more customer-centric and there are calls for the creation of digital platforms allowing adaption to the rapidly changing needs of customers for transport, digital payment, distribution and delivery of personalized goods [7,8].

However, the introduction of new technology may not succeed if employees refuse to use the system or fail to use it in an appropriate way.

Evidence shows that some of the main causes of failure can be traced back to employees’ resistance to change, lack of motivation and other human factors [9,10,11], which can result in work-related stress, dissatisfaction and absenteeism, which are precursors of low job performance.

The literature usually analyzes the impact of factors facilitating the use of technology and the intention to use it, without considering work outcomes in terms of organizational and workers’ wellbeing. In order to bridge this literature gap, our study aimed to combine two theoretical perspectives: one related to the technology acceptance model (TAM), and the other related to workers’ wellbeing. This integration represents the most innovative aspect of this research.

In particular, this study aimed to examine the factors facilitating the acceptance of a new technology by sailors involved in maritime transport in Venice and its impact on intentions to use technology, positive work outcomes (work engagement and work satisfaction) and levels of absenteeism. Absence from work can be considered as a counterproductive behavior [12] that not only impacts the quality of work produced by the employee but also can negatively affect the activities of other employees in the organization in terms of passenger service. In fact, in maritime companies, sailor absenteeism strongly influences the reorganization of work shifts and activities that involve customer service.

To achieve this goal, we focused both on factors strongly related to the perception of technological change and on social support from colleagues. To date, in maritime transport, there has never been any research concerning these particular psychosocial aspects and their influence on sailors’ wellbeing and absenteeism.

Specifically, we assumed that facilitating a constructive attitude towards technology would have a positive impact on intentions to use it. This behaviour proved to influence work satisfaction and to be engaging in motivational processes. In return, these two positive job-related outcomes influenced organizational results in terms of absences from work, which can disrupt work processes, decrease performance and heighten the workload of colleagues.

This research is based on a case of technological innovation in a public maritime transport company, which introduced a new ticket issuing system via a customized device used by workers on board. This was a significant change, because new tasks were assigned to the sailors, thereby increasing their role and responsibilities.

## 2. Literature Review

The study of the factors that facilitate the acceptance and use of new technology can be crucial to the implementation of such technology [13,14,15], in order to predict its actual use by operators and to avoid resistance to change, which often leads to demotivation and a negative attitude towards the technology itself. In this respect, several studies have been conducted in the area of information systems to identify the above-mentioned factors, and many such studies are based on the technology acceptance model (TAM) developed by Davis [16]. A second theoretical framework that we adopted in this study concerns technology as a potential source of stress, by causing burnout, strain and discomfort [17]. In fact, technology could be considered a job demand, requiring extended psychological investment or skills. For example, ICT-related issues and demands include ICT errors, incompatible technologies, expectations of continuous learning, fast responses and constant availability, cognitive overload, and poor quality of communication [18,19]. These potential techno-stressors could lead to exhaustion, especially when organizations do not supply resources at infrastructure and social levels in order to support technological changes [20]. In this sense, the JDR model [21] allows understanding of the motivational process of engagement which is activated when these resources are present, helping workers to have a positive experience of technology and preventing symptoms related to techno-stress.

### 2.1. Theoretical Fundamentals

The technology acceptance model (TAM) developed by Davis [16] rests on the assumptions of the theory of reasoned action (TRA) by Fishbein and Ajzen [22].

The technology assistance model (TAM) suggests that, when people have to use new technology, two main factors—namely its perceived usefulness and ease of use (usability)—influence people’s decision about how and when they will use the technology [16]. Perceived usefulness was defined by Davis [16] as the belief that technology improves workers’ performance. Perceived ease of use was defined as the belief that the technology does not require too much effort [11].

An application of the TAM in the context of shipping can be found in a study by Tsai [2], who examined navigators’ attitudes towards using information systems and devices on board ships. The results indicated that the TAM was able to explain the adoption of information systems in the shipping industry, showing that its ease of use had a significant positive effect on the perceived usefulness of the system and on sailors’ attitudes towards its use.

Another research work [23] attempted to study the factors that influenced passengers’ acceptance and intention to use technological innovations, which involved the use of a device (bracelet) for the traceability and identification of persons on board in the event of a ship evacuation. In terms of utility, the bracelet was considered as an element that would accelerate the evacuation process and an advantage in finding passengers in case of absences at meeting points. The authors’ assumption was that the intention to use a piece of technology represented the main variable capable of predicting the actual use of the device, and the study identified specific factors that influenced such intentions, including expected usefulness, trust, social influence and perceived security risks.

An interesting result exposed the mediational role of perceived usefulness between trust in technology and the intention to use it. A person who doubts the reliability and efficacy of localization systems, and has low confidence in this technology, is likely to distrust, in the event of a disaster, the overall expected usefulness of such localization systems [23]. Other research works have also provided evidence of the positive effect of trust on localization systems [24], thereby confirming the relevance of trust in technology. Up to now, there have been no studies examining the relationships between technology acceptance, well-being and absenteeism in the transport sector.

#### 2.1.1. Work Satisfaction and Engagement: The Approach of the Job Demands–Resources (JD–R) Model

When employees perceive technology as problematic to use and when they do not master technological tools at work, they are more inclined to experience techno-stress. In fact, the implementation of new technology could be considered as a critical period that may have a damaging effect on employees’ physical and mental health, leading to burnout and even to presenteeism [25]. In other words, organizational changes, among which new technology is included, could be defined as a psychosocial risk factor for the appearance of stress symptoms in workers.

However, if workers can count on organizational resources, the introduction of new technology can lead to work satisfaction [26]. In this respect, the job demands–resources (JD–R) model [21] assumes that job resources help employees to manage job demands and, at the same time, make workers learn from and grow in their work activities, with positive consequences in terms of motivation, feelings of achievement, organizational commitment and high performance. Job resources have important outcomes for both employees and organizations, as they influence work well-being and performance by promoting a sense of personal effectiveness and higher levels of engagement and satisfaction [27].

In fact, job resources trigger a motivational process that promotes feelings of fulfilment in employees, thus boosting their work engagement. This is defined as a work-related state of mind that is characterized by vigor, dedication and absorption [28], which in turn has a positive effect on organizational outcomes, such as a decrease in turnover and absenteeism, and an increase in the level of performance.

With regards to technological change, researchers have stressed that effective communication by management, paired with involvement in choices and timing related to technological transformation, as well as adapted training, can be considered resources that positively influence the acceptance and use of new technology, with a strong effect on workers’ well-being [29,30].

Primarily, the literature related to the shipping context has shown that top-management support is an antecedent to the perceived ease of use and usefulness of technology [2], and to the intention to use new technology [29]. Receiving support from superiors and colleagues plays an additional crucial role, because when an organization is very favorable to the introduction of safety innovations, its employees are more likely to perceive the importance of safety regulations; pay more attention to the use of safety procedures; and to have a more positive attitude towards technology [30,31]. Specifically, research conducted by Hu and colleagues [32] revealed that there was a positive relation between support from colleagues and superiors, and perceived ease of use with respect to safety procedures, as well as between organizational support and the perceived utility of the same procedures.

Another study showed that a lack of support among crew members led to unsafe behavior [33]. This body of research emphasizes the importance of social support in stimulating a motivational process that leads to greater commitment in the use of technology by workers and increased work satisfaction.

#### 2.1.2. Aims and Hypotheses

Our research model was based on the previously described literature and included seven observed variables. Two of them were taken from the TAM: perceived usefulness (PU) and perceived ease of use (EOU), with reference to the device being studied. PU is the extent to which a person believes that using a system will enhance his/her performance, while EOU is the extent to which a person believes that using the IT system will be relatively free of effort [34]. Both PU (H1) and EOU (H2) are predictors of the user’s intention to use an IT system and of their subsequent behaviours. In Fishbein and Ajzen’s theory of reasoned action, intention to use is a proximal predictor of behaviour.

Moreover, following the TAM, our research model considered the effect of EOU on PU: an increase in EOU contributed to improved performance; therefore, EOU had a direct impact on PU (H3). Thus, the first three hypotheses are:

**Hypothesis** **1** **(H1).**
*PU is positively and significantly associated with user intentions to use an IT system.*


**Hypothesis** **2** **(H2).**
*EOU is positively and significantly associated with user intentions to use an IT system.*


**Hypothesis** **3** **(H3).**
*EOU is positively and significantly associated with PU.*


Following the model verified by Hu, Griffin and Bertuleit [32], we hypothesised that EOU was positively affected by colleague support (H4). Colleagues can help a sailor with the adoption of a device by providing explanations for the different functions of the equipment, so that it can be seen as more easy to use. If a sailor has an available useful tool for his/her tasks, he/she may become more engaged in his/her job (H5). Furthermore, as the job demands–resources model explains (H6), support from colleagues can positively influence work engagement. Thus, the next hypotheses are:

**Hypothesis** **4** **(H4).***Colleague support is positively and significantly associated with EOU*.

**Hypothesis** **5** **(H5).**
*PU is positively and significantly associated with work engagement.*


**Hypothesis** **6** **(H6).**
*Colleague support is positively and significantly associated with work engagement.*


Job satisfaction, like other work outcomes, can be influenced by the nature of tasks and aspects of work conditions and tools, which IT devices are a part of. Based on research results that showed that the TAM model variables could predict job satisfaction [35], we postulated that, if an employee wanted to adopt the tool that his/her company made available, he/she would be more satisfied with the job than an employee who was unwilling to adopt the tool (H7). Harter et al. [36] demonstrated that the engagement of employees was positively correlated with job satisfaction: a person’s attitude towards their job and, specifically, job satisfaction, could be strengthened by their engagement with work (H8).

As several meta-analyses [37] have shown, job satisfaction and absenteeism are correlated, with a modest, yet significant, and negative relationship with both the duration and the frequency of absences. The ratio indicated that, if a sailor was not satisfied with his/her job, he/she was likely to report a higher level of absence from work (H9). Thus, the last three hypotheses are:

**Hypothesis** **7** **(H7).**
*Intention to use is positively and significantly associated with job satisfaction.*


**Hypothesis** **8** **(H8).**
*Work engagement is positively and significantly associated with job satisfaction.*


**Hypothesis** **9** **(H9).**
*Job satisfaction is negatively and significantly associated with absenteeism.*


All hypotheses are summarized in the model presented in Figure 1.

## 3. Materials and Methods

### 3.1. Participants and Procedure

This study adopted a convenience sample of 122 participants coming from a population of 175 sailors (response rate 70%); 87 of them were males (71%). The average age was 29 years (D = 7.9 years). Seventy percent had a high school diploma, and 23% had bachelor’s degrees.

Sixty-five per cent of the participants had a permanent job contract, while 35% had a temporary employment contract.

The participants had been using the new device, which represents the introduction of the new technology being researched, on average for nine months (SD = 6.6).

Moreover, we had compared the sailors that were research participants (70%) with sailors that did not participate (30%) in the study, by using the chi-square test. There were no significant associations between the categories of sailors and gender, age, education, or type of contract. Thus, the group of participants was substantially homogeneous concerning the other sailors that did not participate in the research for various reasons.

Furthermore, the number of participants can be considered amply sufficient. Bentler and Chou [38] indicated that the number for a reasonable sample, for a path analysis by structural equation models, should be more than 15 times the number of observed variables. On the basis of the role, our model needs a minimum expected sample of 105 participants—so our sample of 122 cases meets the requirements for this type of analysis.

The device, which represented the new introduced technology and the object of this research, supported sailors directly during navigation, checking of tickets, and location of water buses, but also indirectly had an impact on passenger safety in terms of time dedicated to technical activities (mooring the ship, boarding and disembarking passengers) through the use of the device itself. A questionnaire was administered to sailors online directly via the device itself.

The data related to absenteeism were collected through IT management tools that also allowed the collection of data on counterproductive behaviors (i.e., lateness) and company indicators. The objective data were combined with the questionnaire data for each participant. Data were made anonymous before providing it to the research group. Ethics approval was not required for this study, since the data did not include medical aspects and were treated anonymously.

### 3.2. Measures

The questionnaire included 6 scales. The items of technology acceptance were based as far as possible on validated models such as TAM [16] and TAM 3 [39]; all items related to dimensions of TAM adopted a five-point scale ranging from “1” (strongly disagree) to “5” (strongly agree).

*Perceived Usefulness* (PU) was analysed by five items (e.g., “Using the device would improve my job performance”). In the present study, a confirmatory factor analysis (CFA) verified the mono-dimensionality of the scale (CFI = 0.97; RMSEA = 0.07). The coefficient alpha showed a very high score (0.97), meaning that some items could be redundant [40].

Six items were adopted for measuring perceived ease of use (e.g., “I would find it easy to get the device to do what I want it to do”). In the present study, a CFA confirmed a mono-factor model (CFI = 0.99; RMSEA = 0.06), and the coefficient alpha showed a very high score (0.93).

To measure *intention to use*, we adopted three items (e.g., “Given that I had access to the device, I predict that I would use it”). Since the scale was composed of three items, CFA was not performed; nevertheless, the alpha coefficient showed a good score (0.86).

*Support of colleagues* was assessed through four items of the job content instrument [41,42] (e.g., “The people I work with help me to get the job done”). The items adopted a five-point scale ranging from “1” (strongly disagree) to “5” (strongly agree). In the present study, the CFA showed a good fit of the data with a mono-factor model (CFI = 0.99; RMSEA = 0.01). The alpha index was 0.76.

An Italian version of the nine-item Utrecht work engagement scale [43,44] was adopted to measure *work engagement*. Even if the items were grouped into three subscales, the recommendation of Schaufeli et al. [43] was followed and an overall engagement score of the UWES was calculated, which was used in the analyses. All items (e.g., “I am enthusiastic about my job”) were scored on a seven-point scale ranging from “1” (never) to “7” (always). In the present study, the CFA showed a good fit of the data with a mono-factor model (CFI = 0.98; RMSEA = 0.07). The alpha index of Cronbach was 0.92.

Job satisfaction was assessed with a single item [45]. The item was “Overall, how satisfied are you with your job?” and was scored on a five-point scale ranging from “1” (not satisfied at all) to “5” (completely satisfied).

Finally, the HR department provided yearly absenteeism data (the sum of days of absence from work), based on personnel records, covering the three months after the data collection by the questionnaire. As is typical, the absenteeism data were positively skewed; thus, we applied a square root transformation to approximate normality [46].

### 3.3. Data Analysis

Means, standard deviations, and correlations among the study variables were computed as preliminary analyses, by IBM SPSS version 25. To evaluate the psychometric properties of the measures, a reliability analysis was performed by IBM SPSS Statistics (version 25), and measurement of the CFA was assessed for all measures by IBM AMOS (version 22).

Structural equation models were used to test the model of Figure 1 using IBM AMOS (version 22). The asymptotically distribution-free method (ADF) was adopted, since data screening showed deviations from normality (e.g., kurtosis and skewness). Hair, Black, Rabin, and Anderson [47] recommend the use of at least one fitness index from each category of model fit. Thus, RMSEA (root mean square of error approximation) to test absolute fit, and CFI (comparative fit index) and TLI (Tucker–Lewis index) for the incremental fit were adopted. In the end, we wanted to analyse the parsimony of the model based on Chisq/df (the ratio of the model χ^2^ and the degrees of freedom). Schweizer [48] summarises the acceptable levels of fit indexes: RMSEA below 0.08; the CFI and TLI values should be in the range of 0.90 to 1.00; normed χ^2^ is supposed to stay below 3.

As the variables were collected at two distinct times and the questionnaire contains reversed items [49], we believe that the concerns about common method bias are not a threat to our analyses.

We accepted the hypotheses that presented a *p*-value of less than 0.01.

## 4. Results

### 4.1. Descriptive Statistics

Means, standard deviations, Cronbach’s alpha and correlations are presented in Table 1. In general, the findings show that job satisfaction and absenteeism do not have a normal distribution on the basis of the kurtosis index.

Cronbach alpha’s indexes show a fear range (from 0.76 to 0.97). The correlation matrix presents a good range of values (from −0.22 to 0.72), suggesting that common method bias is not of concern in our data. Moreover, the highest correlation index was between PU and Intention to use (r = 0.72); the lowest correlation indexes regarded absenteeism with respect to PU (r = 0.00) and EOU (r = 0.00). The model’s dependent variable, absenteeism, was correlated only with job satisfaction (r = −0.22).

### 4.2. Main Analyses

The hypothesised model showed that fit indexes were at a good level: Chi square = 13.71; DF = 12; Chi square/DF = 1.14; TLI = 0.96; CFI = 0.98; RMSEA = 0.03 (Figure 2). Every hypothesised standardised regression weight was significant at least to an alpha level of 0.01; the specific direct, indirect and total effects are shown in Table 2.

The variance explained by the model was 4% for absenteeism, 62% for job satisfaction, 33% for work engagement and 58% for intention to use the device.

The results supported all hypothesized relationships, with a statistical confidence level equal to, or lower than, 0.01 (Table 2).

The intention to use the device was significantly and positively affected by perceived ease of use (H1; β = 0.30, *p* < 0.001) and perceived usefulness (H2; β = 0.54, *p* < 0.001). Perceived ease of use was significantly and positively related to perceived usefulness (H3; β = 0.63, *p* < 0.001) and support from colleagues (H4; β = 0.34, *p* < 0.001). Work engagement was significantly and positively related to perceived usefulness (H5; β = 0.29, *p* < 0.001) and support from colleagues (H6; β = 0.43, *p* < 0.001). Job satisfaction was significantly and positively affected by intention to use the device (H7; β = 0.24, *p* < 0.01) and work engagement (H8; β = 0.68, *p* < 0.001). Finally, the results showed that job satisfaction predicted, negatively, the absenteeism data that were measured three months later (H9; β = 0.20, *p* < 0.01). Thus, the higher direct effects concerned the influence of the intention to use on job satisfaction and impact of EOU on PU. Lower significant direct effects were job satisfaction on absenteeism and work engagement on job satisfaction. The higher indirect effects concerned the relationship between the support from colleagues and job satisfaction, and the relationship between EOU and intention to use.

The variance explained by the model was 4% for absenteeism, 62% for job satisfaction, 33% for work engagement, 58% for intention to use the device, 40% for PU and 12% for EOU.

## 5. Discussion

The current research aimed to provide more in-depth insight into the process of acceptance of a new technology in the context of shipping and the impact of such technology on workers’ well-being and absenteeism through the integration of two theoretical perspectives related to the TAM and the JD–R models.

On the basis of the results, all hypotheses (from H1 to H9) have been accepted.

According to our hypothesis, perceived ease of use (H3) predicted perceived usefulness; both were related to the intention to use the device of sailors (H1 and H2), with a positive impact on work satisfaction (H7). In accordance with these results, a recent meta-analysis [50] confirmed that the TAM successfully predicted user behavior. This model was also adopted to explain drivers’ acceptance of driver support systems in transport [51]. Perceived usefulness was founded on a belief in increased performance with the new technological tools [16]. All this means that, in the context of acceptance by sailors, the usefulness of a device and interaction with it need to be well communicated, so as to create a positive effect on the use of the technology. The benefits of increased performance should motivate workers to use new technologies.

In line with other studies that have examined the role of job-related well-being in technology acceptance [51], our research showed that the perceived usefulness of the device influenced not only the intention to use the technology, but also the engagement of workers (H5). As a result, work satisfaction increased, which, in turn, prevented the potential symptoms related to techno-stressors and reduced the level of absenteeism. This result suggests that perceived usefulness and its subsequent effects may play a crucial role in defining overall user experience evaluations by also preventing psychological symptoms such as anxiety, fatigue and skepticism, which other studies showed to be related to techno-stress [52,53]. Other researches pointed out how perceived usefulness had a significant relationship with experienced valence in successful technology adoptions [54,55].

With the increase in automation, the introduction of technological innovations in the shipping industry has led to a reduction of personnel, with resultant changes in maritime roles—particularly in terms of constant monitoring of systems like radars and GPSs. Although these instruments can replace some of the workforce, they do relieve workers from a lot of mental fatigue, which consequently impacts on stress, health, teamwork, communication and safety culture [56]. In this sense, the perceived usefulness examined in our work activated the intention to get involved in the use of new technology and also acted as a protective factor with respect to the onset of stress symptoms, thereby enhancing the motivational process of work engagement.

According to the recent literature [57], work engagement translates into seeing the technological change as a challenge full of meaning, and technology change is perceived as something that calls for resiliency and perseverance and therefore positively influencing worker satisfaction (H8), whilst decreasing levels of absenteeism. The level of engagement at work is important, because it indicates how interesting workers truly find their job. Organizations that invest in supporting workers during technological changes favour workers’ willingness to dedicate their efforts and abilities to the work task.

In this respect, the introduction of new technology that increases work engagement could be an indicator of the positive impact that the technology brings with it.

The second main result had to do with the influence of social support, not only on perceived ease of use (H4), which was in line with the evidence provided by Hu and colleagues’ research [32], but also on worker engagement (H6), which in turn was a predictor of job satisfaction (H8)—thereby decreasing absenteeism (H9). Based on the JD–R model used in this study, work engagement was also predicted by us as a result of the support received from colleagues, which represented a resource that could buffer the adverse consequences of technological demands and could hence offer support to secure high levels of engagement and subsequent positive outcomes [58,59]. An organizational context that is perceived as supportive favours the ability of workers to focus their efforts and abilities on work tasks.

In line with other research [60], the results of our own imply that the intention to use new technology, when this is perceived as positive and non-threatening by workers, is a precondition to work satisfaction (H7) and will decrease negative work outcomes, such as the intention to leave and the level of absenteeism, measured in this study by objective indexes four months after the administration of the questionnaire.

Similarly, support from colleagues seems to have a pivotal role in increasing the likelihood of workers of being successful in achieving their professional goals and obtaining job satisfaction, in turn preventing fatigue and stress symptoms, and reducing levels of absenteeism in particular. Other studies too pointed at social support as an important job resource in the context of maritime transport: a supportive social climate onboard ships can indeed positively influence well-being and performance [61,62].

Absenteeism, classified as a counterproductive behavior, and a specific and negative component of individual performance [63], represents a very critical element in human resource management—especially for a public transportation company, as it strongly affects customer service both in the short and in the long term. The two-wave design of this research allowed for the identification of aspects associated with the acceptance of new technology and the motivational processes of work engagement and satisfaction that could predict long-term objective indicators of absenteeism [64].

Another fundamental aspect of the introduction of technology in public transport is its direct impact on customers. There is general consensus that the introduction of a driver support system can indeed improve customer service and transportation safety, reduce the perceived waiting time, and increase the sense of security and value for money [64]. The device examined in this paper may be included in information systems that do not actively intervene in the driving task of water buses, although it supports pilots and sailors by providing information, warnings, geographic locations and sails—thus ensuring better customer service. From the point of view of practical implications, a study of the aspects that promote the correct use of technological tools could have an impact on public transport services offered in cities. The results would have a significant value to the whole management of company resources, by impacting the quality of public services connected with Venetian mobility.

Several companies have started adopting a socio-technical approach to the introduction of technological innovations, not only by focusing on technical and engineering aspects or on computer science, but also by considering the social implications of technological change. Failing to consider the human factor in a technical system will have an impact on the system’s performance and its ability to function safely. All of that may mean reshaping workflows and roles to allow for maximum autonomy, control and decision making for employees. Indeed, organizations that have a strong innovation policy and a user-guided implementation strategy are generally more successful [65,66]. They are characterized by a plan of efficient business, top-management support for innovation and systematic methods of managing technological change. This last point requires the involvement of workers in terms of good training, technical assistance and high-quality communication, in order to favour their perception of ease of use and usefulness. A social campaign may be used to help workers better understand the benefits of adopting new technologies and to create social acceptance among them.

Our results confirm, in fact, that if workers perceive that their efforts and workload are manageable, or that there are additional resources to allocate, the workers will most likely accept the use of a new technological device.

### Limitations

This study, however, has several limitations. Firstly, the study could have included other variables related to the acceptance of a new technology, such as hedonic motivation and habits in accepting new technologies. Secondly, the role of sailors was not examined exhaustively. It is recommended that performance indicators, such as task performance, customer satisfaction in terms of fewer complaints (conflicts), errors in maritime activities (security), and accounting reports, be examined in future research. Thirdly, not all sailors involved in the program participated in the research. However, the characteristics of the uninvolved workers did not differ from those who participated in the research. Finally, in addition to social support from colleagues, other variables related to the crew onboard and their relationships with the pilot could also be included.

## 6. Conclusions

The importance of this research mainly lies in the knowledge that transport companies can gain regarding which aspects they should focus on when implementing technology and on how to help their workers use it. Technological change has forced many workers to adapt and people are obliged to learn new tasks, including how to use hi-tech devices and work with data given by a machine, which become the most significant factors of their activities at work. However, in companies, the focus has often been on the technology itself, and less attention has been paid to the technology users, in connection with their acceptance and utilization.

This study examined how different factors influenced workers’ intentions to use new technology. In this regard, the first implication of this research was related to the role of perceived ease of use and usefulness, as a determinant of the willingness to use IT. The results showed that workers needed to perceive themselves as being able to be part of the system and to understand the technological change, as well as to feel like they could somewhat master the use of the technology. Another crucial psychosocial aspect was related to social support. Worker feedback and involvement, with the support from their colleagues who share and face the same changes every day, made workers more aware of the positive impact of technological innovations. This made workers more likely to see technological change as an opportunity, raising positive expectations towards the use of new tools.

Transport companies, therefore, should use this methodology of involving users by clearly communicating about the implementation of new technologies and focusing their training on the development of skills that can be perceived as an increase and enrichment of the roles of sailors.

The public transportation company examined in this work had involved, on a voluntary basis, a work team of sailors, who directly followed and supported the implementation of a new device, and who were allowed to give suggestions that could improve the functioning of the device. The graphic aspects of the technological tool were refined, so as to give greater visibility of its icons and improve their intuitiveness; the group further contributed by giving indications on how to tailor the contents and optimize the installed apps, thereby improving the performance of the tool. Finally, the activation of a help desk, which also operated through social channels (messages on telegram) for the immediate resolution of problems, as well as the supply of a printer connected to the device, represented facilitating conditions for the perceived ease of use of the technology.

These aspects related to the acceptance of new technology were found to be associated with the motivational process of engagement and satisfaction that predicted long-term absenteeism. It is therefore important to pursue these aspects to improve this type of performance.

Additionally, direct and positive relationships with superiors and colleagues represented drivers of motivational processes that, allowing workers to open up and be ready for technological change, consequently also impacted on their own well-being and performance.

In terms of practical implications, the involvement of the team and direct superiors in the implementation of the new technology were key elements for future innovation in the transport company examined, as well as having a direct impact on public and customer service.

From a coaching point of view, the training of specific profiles not only on technical skills, but also prepared on aspects related to the management of organizational changes, could be successful in terms of performance and worker satisfaction.

## Figures and Tables

**Figure 1 ijerph-18-12358-f001:**
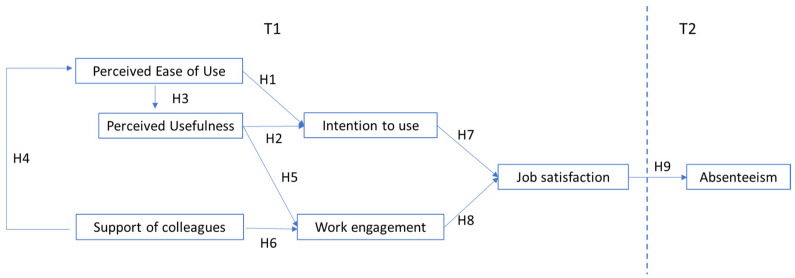
The hypothesized model.

**Figure 2 ijerph-18-12358-f002:**
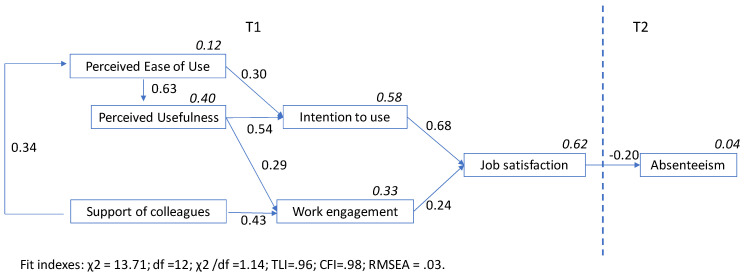
The tested model (*N* = 122). Note: Every effect is statistically significant at *p* < 0.01; italicized numbers present the squared multiple correlations.

**Table 1 ijerph-18-12358-t001:** Means, standard deviations and correlations among study variables (*N* = 122).

Variable	M	SD	Skew	Kurt	1	2	3	4	5	6
1. Perceived usefulness (PU)	3.33	1.06	−0.23	−0.71	(0.97)					
2. Perceived ease of use (EOU)	3.43	0.90	−0.39	0.15	0.62 ^**^	(0.93)				
3. Intention to use	2.89	1.03	0.11	−0.29	0.72 ^**^	0.65 ^**^	(0.86)			
4. Support from colleagues	3.56	0.71	−0.56	1.53	0.20 ^*^	0.32 ^**^	0.26 ^**^	(0.76)		
5. Work engagement	5.18	1.46	−0.84	0.09	0.39 ^**^	0.37 ^**^	0.37 ^**^	0.46 ^**^	(0.92)	
6. Job satisfaction ^^*^^	5.41	1.56	−1.21	1.01	0.42 ^**^	0.32 ^**^	0.43 ^**^	0.44 ^**^	0.73 ^**^	--
7. Absenteeism ^^*^^	6.75	2.11	−0.97	3.53	0.00	0.00	−0.04	−0.15	−0.15	−0.22 ^*^

Note. Cronbach’s alphas for the sample (in parentheses) can be found on the diagonal; ^**^
*p* < 0.01; ^*^
*p* < 0.05.

**Table 2 ijerph-18-12358-t002:** Standardised effects (*N* = 122).

	Dependent Variables					
Direct Effects	Perceived Ease of Use	Perceived Usefulness	Intention to Use	Work Engagement	Job Satisfaction	Absenteeism
Perceived ease of use		0.63 ^***^ (H3)	0.30 ^***^ (H1)			
Perceived usefulness			0.54 ^***^ (H2)	0.29 ^***^ (H5)		
Intention to use					0.24 ^**^ (H7)	
Support from colleagues	0.34 ^***^ (H4)			0.43 ^***^ (H6)		
Work engagement					0.68 ^***^ (H8)	
Job satisfaction						−0.20 ^**^ (H9)
Absenteeism						
Indirect Effects						
Perceived ease of use			0.34 ^***^	0.18 ^*^	0.28 ^***^	−0.06
Perceived usefulness					0.33 ^***^	−0.07
Intention to use						−0.05
Support from colleagues		0.22 ^**^	0.22 ^**^	0.06	0.39 ^***^	−0.08
Work engagement						−0.14
Job satisfaction						
Absenteeism						
Total Effects						
Perceived ease of use		0.63 ^***^	0.64 ^***^	0.18 ^*^	0.28 ^***^	−0.06
Perceived usefulness			0.54 ^***^	0.29 ^***^	0.33 ^***^	−0.07
Intention to use					0.24 ^**^	−0.05
Support from colleagues	0.34 ^***^	0.22 ^**^	0.22 ^**^	0.50 ^***^	0.39 ^***^	−0.08
Work engagement					0.68 ^***^	−0.14
Job satisfaction						−0.20 ^**^
Absenteeism						
Squared multiple Correlations (R squared)	0.12	0.40	0.58	0.33	0.62	0.04

Note. ^***^
*p* < 0.001; ^**^
*p* < 0.01; ^*^
*p* < 0.05.

## Data Availability

The data presented in this study are available on request from the corresponding author.

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
