# Peer review of "The Predictive Factors of New Technology Adoption, Workers’ Well-Being and Absenteeism: The Case of a Public Maritime Company in Venice"

_ijerph, 2021, doi:10.3390/ijerph182312358_

Round 1

Reviewer 1 Report

Dear Authors, thank You for so interesting research.

The article is written on the relevant topic and is well structured as well as logically proved.

However I'd recommend making some improvements to the structure of the article:

  1. Please kindly shorten the section 1 Introduction setting aside the literature review and help the reader understanding the following important issues: the research gap which the authors could disclose in the article; the relevance of the topic; the research question and the aim of the article. All those elements could be described on one page;
  2. The second section could be entitled as 2. Materials and method with the following subsections: 2.1. Literature review with approximately 25 references being moved from the Introduction; 2.2. Theoretical fundamentals with 2.2.1 Work satisfaction and engagement: the approach of the Job Demands-Resources (JD-R) model. and 2.2.2. Studies Hypotheses. 2.3. Participants. 2.4. The device. 2.5. Procedure. 2.6. Instruments. 2.7. Data analysis 
  3. Results section.
  4. Discussion section containing the limitations of the model and the issues for future research. The greatest part of the discussion relating to the steps of the model could be moved to the Results section.
  5. Conclusion. All materials remaining from the Discussion section after describing the limitations of the model and the issues for future research.

It could be recommended to add to the literature review some significant references regarding the digitalization in logistics and retail:

Barykin, S.Y.; Kapustina, I.V.; Sergeev, S.M.; Kalinina, O.V.; Vilken, V.V.; Putikhin, Y.Y.; Volkova, L.V. Developing the physical distribution digital twin model within the trade network. Academy of Strategic Management Journal 2021, 20, 1–24.

Barykin, S.Y.; Bochkarev, A.A.; Dobronravin, E.; Sergeev, S.M. The place and role of digital twin in supply chain management. Academy of Strategic Management Journal 2021, 20, 1–19.

I'd recommend the acceptance of the article after this changes.

Author Response

Dear reviewer,

we would like to thank you for your conscientiousness in reviewing our manuscript. We hope that the following replies to your comments can address the issues you have raised. All the changes are marked.

The Authors

REPLIES TO REVIEWER

  1. Please kindly shorten the section 1 Introduction setting aside the literature review and help the reader understanding the following important issues: the research gap which the authors could disclose in the article; the relevance of the topic; the research question and the aim of the article. All those elements could be described on one page;It could be recommended to add to the literature review some significant references regarding the digitalization in logistics and retail:

Barykin, S.Y.; Kapustina, I.V.; Sergeev, S.M.; Kalinina, O.V.; Vilken, V.V.; Putikhin, Y.Y.; Volkova, L.V. Developing the physical distribution digital twin model within the trade network. Academy of Strategic Management Journal 2021, 20, 1–24.

Barykin, S.Y.; Bochkarev, A.A.; Dobronravin, E.; Sergeev, S.M. The place and role of digital twin in supply chain management. Academy of Strategic Management Journal 2021, 20, 1–19.

I'd recommend the acceptance of the article after this changes.

We added in the Introduction the research gap and aims of the article (59-77), explaining that the literature usually analyses the impact of factors facilitating the use of technology on the intention to use it, without considering work outcomes in terms of organizational and workers’ wellbeing. In order to bridge this literature gap, our study aimed to combine two theoretical perspectives, one related to the technology acceptance model (TAM) and one related to workers’ wellbeing. This integration represented the most innovative aspect of this research.

Particularly, this study aimed to examine the factors facilitating the acceptance of new technology by the sailors of the maritime transport of Venice and their impact on the intention of use technology, positive work outcomes (work engagement and work satisfaction), and level of absenteeism. The absence from work can be considered as an indirect job performance indicator (Truxillo, 2016) related to passengers’ service. In fact, in maritime company sailors’ absenteeism strongly influences the reorganization of work shifts and activities that involve customer service. To achieve this goal, we focused both on factors more related to the perception of technological change and on the social support of colleagues. To date, there is no research concerning these psychosocial aspects in the maritime transport sector and their influence on sailors’ wellbeing and absenteeism.

Furthermore, we added the references you have suggested:

Barykin, S.Y.; Kapustina, I.V.; Sergeev, S.M.; Kalinina, O.V.; Vilken, V.V.; Putikhin, Y.Y.; Volkova, L.V. Developing the physical distribution digital twin model within the trade network. Academy of Strategic Management Journal 202120, 1–24.

Barykin, S.Y.; Bochkarev, A.A.; Dobronravin, E.; Sergeev, S.M. The place and role of digital twin in supply chain management. Academy of Strategic Management Journal 202120, 1–19.

  1. The second section could be entitled as 2. Materials and method with the following subsections: 2.1. Literature review with approximately 25 references being moved from the Introduction; 2.2. Theoretical fundamentals with 2.2.1 Work satisfaction and engagement: the approach of the Job Demands-Resources (JD-R) model. and 2.2.2. Studies Hypotheses. 2.3. Participants. 2.4. The device. 2.5. Procedure. 2.6. Instruments. 2.7. Data analysis 

Thank you. As you suggested, we have changed the titles of the  paragraphs.

  1. Results section.

The Results section was revised on the basis of the hypotheses. The results have been described in more detail.

  1. Discussion section containing the limitations of the model and the issues for future research. The greatest part of the Discussion relating to the steps of the model could be moved to the Results section.

A Part of the Discussion was moved to conclusions.

  1. All materials remaining from the Discussion section after describing the limitations of the model and the issues for future research.

A Part of the Discussion was moved to conclusions and a paragraph related to limitations has been added.

Reviewer 2 Report

The article entitled: "The importance of psychosocial factors in the adoption of new technology for shipping transport: the case of the water buses of a public maritime company in Venice" requires several adjustments. I present to the author of the article the most important comments that need to be made for publication.

The most important shortcoming of the article is the overall understanding of performance throughout the article. In some parts of the article, I have the impression that the authors consider performance to be equal to attendance. Other comments and recommendations are:

  • the title of the article does not match the content, I recommend the authors to modify it to reflect the purpose of the article.
  • Abstract, lines 11, 17 and 19, remove the part names: (Background, Methods and Results).
  • Introduction: I recommend the authors to supplement the literature review with a performance that is completely absent in view of the stated aim. The authors should replace (and 415 million passengers in Europe alone [4].) With up-to-date information, source 4 from 2019, which I consider inappropriate given the topicality. Line 46: of motivation and other human factors [6-7-8]. The source should be listed as follows: [6-8]. Line 99: individual performance, how can this term be understood? Again, with regard to the issues addressed, it is necessary to pay more attention to the definition of the term. Line 146: 1.3. Studies Hypotheses is the weakest part of the first part of the paper! Authors must introduce Figure 1 into the text. The bigger problem is that the authors do not give an explicit formulation of hypotheses (H1 to H9). It is necessary for the authors to formulate all hypotheses, for example: This is an example of a null hypothesis: Employee candidates who have a minimum of four years of work experience are more likely to receive an interview. no work experience receive as many interview invitations as those with a minimum of four years.). Characteristics of a good hypothesis: conceptually clear, specificity, testability, availability of techniques, theoretical relevance, consistency, objectivity and simplicity.
  • Materials and Methods, is the sample sufficiently representative (the participants were 122 sailors)? It is debatable whether the sample is sufficient. Personally, does the sample seem too small to me? What is the base file when the authors present a sample (122 participants). It is necessary to expand the whole part (2.1. Participants)! Line 192-193 (The data related to absenteeism were collected through IT management tools that allow to collect data on performance and company indicators.) It is necessary to explain what is "collect data on performance". Line 233: Different analyses were conducted - please remove!
  • Results: I recommend that the authors move 3.1. Preliminary analysis for Material and Methos. Lines 274-284 need to be reworked (see comments on hypothesis formulation). The above assessment is inappropriate.
  • Discussion, line 352 (... well-being and performance [55-56], should be [55, 56] or [55 and 56]. The discussion is interesting, but given the vague evaluation of the hypotheses, the discussion is debatable. Line 371-373 (Overall, the importance of this research mainly lies in the knowledge that transport companies can gain on the aspects they should focus on when implementing technology and on how to help their workers use it.) line 400-407 needs to be moved to the conclusions.
  • The conclusions are vague and repetitive with regard to the Discussion. I recommend the authors to state explicit results and draw clear conclusions from them.

Overall assessment: The presented article is interesting, but it contains several shortcomings, whether methodological or logical. I encourage the authors to incorporate the proposed changes that may help to improve the article. I wish the authors much success in their work.

Author Response

Dear reviewer,

we would like to thank you for your conscientiousness in reviewing our manuscript. We hope that the following replies to your comments can address the issues you have raised. All the changes are marked.

The Authors

REPLIES:

  1. The title of the article does not match the content, I recommend the authors to modify it to reflect the purpose of the article.

Yes, as you suggested, we have changed the title, reducing the length.

  1. Abstract, lines 11, 17 and 19, remove the part names: (Background, Methods and Results).

Thank you, we removed in the abstract the part names: Background, Methods and Results.

  1. Introduction: I recommend the authors to supplement the literature review with a performance that is completely absent in view of the stated aim. Line 99: individual performance, how can this term be understood? Again, with regard to the issues addressed, it is necessary to pay more attention to the definition of the term.

Thank you, for the advise. Firstly, concerning the review related to performance, we underlined in the text that absenteeism can be considered a counterproductive behaviour that does not only impact the quality of work produced by the employee but also can negatively affect the activities of other employees in the organization and create undesirable risks for the employer. In the specific case of the transport sector, absenteeism has a very strong impact on customer service, which represents the main objective of economic effectiveness.

Secondly, We decided to focus more on absenteeism in the text than individual performance.

  1. The authors should replace (and 415 million passengers in Europe alone [4].) With up-to-date information, source 4 from 2019, which I consider inappropriate given the topicality.

Thank you for your suggestion. We added a reference to the situation of the sector of maritime transport in the post-pandemic era.

  1. Line 46: of motivation and other human factors [6-7-8]. The source should be listed as follows: [6-8].

The source [6-8] has been listed as you suggested.

  1. Line 146: 1.3. Studies Hypotheses is the weakest part of the first part of the paper! Authors must introduce Figure 1 into the text. The bigger problem is that the authors do not give an explicit formulation of hypotheses (H1 to H9). It is necessary for the authors to formulate all hypotheses, for example: This is an example of a null hypothesis: Employee candidates who have a minimum of four years of work experience are more likely to receive an interview. no work experience receives as many interview invitations as those with a minimum of four years.). Characteristics of a good hypothesis: conceptually clear, specificity, testability, availability of techniques, theoretical relevance, consistency, objectivity and simplicity.

Thank you for this important advice. The hypotheses have been completely reformulated and clearly specified, following some papers of the same Special Issue. The figure 1 is inserted in pag. 5.

  1. Materials and Methods, is the sample sufficiently representative (the participants were 122 sailors)? It is debatable whether the sample is sufficient. Personally, does the sample seem too small to me? What is the base file when the authors present a sample (122 participants). It is necessary to expand the whole part (2.1. Participants)!

Thank you, this is a significant question that can improve our paper. We have adopted a convenience sample that is the 70% of the population. There are no statistically significant differences between participants and non-participants regarding gender, age, education and work contract. The percentage of participants is not high but the number of the sample is sufficient to satisfy the necessary requirements for the structural equation models adopted in the statistical analyzes. On the basis of the Bentler and Chou [39] role, our model needs a minimum expected sample of 105 participants; so, our sample, of 122 cases, has the requirements for this type of analysis. We have added information on all that.

Anyway, if you want a parameter from Power analysis: the structural equation model is based on a correlation matrix. So, for a correlation index, if we adopt a D=0.3, an Alpha of 0.05 (with a tail), a Power of .80, the sample should be 64 participants (almost half of our sample).

  1. Line 192-193 (The data related to absenteeism were collected through IT management tools that allow to collect data on performance and company indicators.) It is necessary to explain what is "collect data on performance".

The IT management tools of the company collect and manage some data on counterproductive behaviors as absenteeism, tardiness, and lateness. Thank you, we have improved the sentence

  1. Line 233: Different analyses were conducted - please remove!

Thank you, we have deleted the sentence.

  1. Results: I recommend that the authors move 3.1. Preliminary analysis for Material and Methos.

Thank you, for the advice. We have moved the text, and we have renamed the first section of the result as “Descriptive Statistics,” following some papers of the same Special Issue.

  1. Lines 274-284 need to be reworked (see comments on hypothesis formulation). The above assessment is inappropriate.

Thank you for having underlined all that. Now, in the Result section, we explain if the evidence supports the hypotheses and if they are confirmed or not.

  1. Discussion, line 352 (... well-being and performance [55-56], should be [55, 56] or [55 and 56]. The Discussion is interesting, but given the vague evaluation of the hypotheses, the Discussion is debatable. The reference list (55 and 56) have been corrected. In the discussion the hypotheses have been recalled.
  2. Line 371-373 (Overall, the importance of this research mainly lies in the knowledge that transport companies can gain on the aspects they should focus on when implementing technology and on how to help their workers use it.) line 400-407 needs to be moved to the conclusions.

As you suggested, the phrase (Overall, the importance of this research mainly lies in the knowledge that transport companies can gain on the aspects they should focus on when implementing technology and on how to help their workers use it) has been moved to the conclusions.

Also the part related to the process of implementation by the public transportation company (The public transportation company examined in this work had involved, on a voluntary basis, a work team of sailors, who directly followed and supported the implementation of a new device, and who were allowed to give suggestions that could improve the functioning of the device. The graphic aspects of the technological tool were refined, so as to give greater visibility to its icons and improve their intuitiveness; the group further contributed by giving indications on how to tailor the contents and optimize the installed apps, thereby improving the performance of the tool. Finally, the activation of a help desk, which also operated through social channels (messages on telegram) for the immediate resolution of problems, as well as the supply of a printer connected to the device, represented facilitating conditions for the perceived ease of use of the technology) has been moved to the conclusions.

  1. The conclusions are vague and repetitive with regard to the Discussion. I recommend the authors to state explicit results and draw clear conclusions from them.

Thank you for the suggestion. The results were better explained, and the practical implications for each result were presented.

Reviewer 3 Report

Thank you for the opportunity to review this manuscript.

The article deals with the adoption of new technology for shipping transport.

After carefully reviewing this manuscript, I formulated suggestions to support the further improvement of the manuscript.

Hypotheses should be formulated clearly and exactly.

I think that the reason for carrying out the research and the gap in the science should be reformulated and better revealed.

Parts 2.1, 2.2 and 2.3 are too short to be numbered separately.

It is not clear from the paper whether data on absences were provided for a sample of respondents or for sailors in general.

The title of the paper is “The importance of psychosocial factors in the adoption of new technology for shipping transport…” but in the paper I did not find any psychosocial factors that would be examined. The authors should define and determine what they consider to be psychosocial factors, resp. psychosocial factors of work.

The presented results should be described in more detail. The reasons why authors not refute hypotheses should also be more precisely justified.

The authors state at the lines 290-293: “ The current research aimed to provide more in-depth insight into the process of acceptance of a new technology in the context of shipping and the impact of such technology on workers’ well-being and performance through the integration of two theoretical perspectives related to the TAM and the JD-R models.” How was measured the impact of such technology on workers’ well-being and performance? Which indicators connected to well-being and performance were monitored or measured?  How was examined the role of job-related well-being in technology acceptance (lines 304-305)?

The authors claim that “… work satisfaction increased…“ and „…which, in turn, prevented symptoms of strain and reduced the level of absenteeism.“  (lines 307-308). What is this claim based on? What symptoms of strain the authors of the paper investigated? Did they monitor changes in the satisfaction and absenteeism over time?

The authors state in the conclusion: “This study examined how different social factors influenced workers’ intention to use new technology. In this regard, the first implication of this research was related to the role of perceived ease of use and usefulness, as a determinant of the willingness to use IT.” So, which social factors were examined?

In conclusion, there are several statements that are not the result of the presented research, e.g. “Workers’ involvement, and their awareness of the impact of technology on performance effectiveness and its efficiency, fed positive expectations towards the use of new tools.” The most important findings of the presented research should be summarized in the conclusion.

Author Response

Dear reviewer,

we would like to thank you for your conscientiousness in reviewing our manuscript. We hope that the following replies to your comments can address the issues you have raised. All the changes are marked.

The Authors

REPLIES:

  1. Hypotheses should be formulated clearly and exactly.

Thank you for underlying the standard of the journal. Now the hypotheses are clearly written.

  1. I think that the reason for carrying out the research and the gap in the science should be reformulated and better revealed.

Thank you, for this important suggestion. We added in the introduction the research gap and aims of the article (59-77), explaining that the literature usually analyses the impact of factors facilitating the use of technology on the intention to use it, without considering work outcomes in terms of organizational and workers’ wellbeing. In order to bridge this literature gap, our study aimed to combine two theoretical perspectives, one related to the technology acceptance model (TAM) and one related to workers’ wellbeing. This integration represented the most innovative aspect of this research.

Particularly, this study aimed to examine the factors facilitating the acceptance of a new technology by the sailors of the maritime transport of Venice and their impact on the intention of use technology, positive work outcomes (work engagement and work satisfaction), and level of absenteeism. The absence from work can be considered as an indirect job performance indicator (Truxillo, 2016) related to passengers’ service. In fact, in maritime company sailors’ absenteeism strongly influences the reorganization of work shifts and activities that involve customer service. To achieve this goal, we focused both on factors more related to the perception of technological change and on the social support of colleagues. To date, there is no research concerning these psychosocial aspects in the maritime transport sector and their influence on sailors’ wellbeing and absenteeism.

  1. Parts 2.1, 2.2 and 2.3 are too short to be numbered separately.

We have changed the numbering, as follow:

  1. Materials and Method

3.1 Participants and Procedure

3.2 Measures

3.3 Data analysis

  1. It is not clear from the paper whether data on absences were provided for a sample of respondents or for sailors in general.

Thank you for the suggestion. The research adopted a sample of sailors, now we have explained with more details in the Participants section. Moreover, the number of absences were computed for any participant (as at individual level): 122 on a total of 174 sailors (70%)

  1. The title of the paper is “The importance of psychosocial factors in the adoption of new technology for shipping transport…” but in the paper I did not find any psychosocial factors that would be examined. The authors should define and determine what they consider to be psychosocial factors, resp. psychosocial factors of work.

Thank you, for the suggestion. We have changed the title.

In the text, we have underlined that we focused on psychosocial factors, referring both to variables concerning the subjective perception of technology and psychosocial variables in terms of social support of colleagues (pp. 2, line 68-71)

  1. The presented results should be described in more detail. The reasons why authors not refute hypotheses should also be more precisely justified.

Thank you for the advice. We have added details regarding both descriptive analysis and path models. We have added a sentence in the Data analysis section that describes the role of accepting the hypotheses.

  1. The authors state at the lines 290-293: “ The current research aimed to provide more in-depth insight into the process of acceptance of a new technology in the context of shipping and the impact of such technology on workers’ wellbeing and performance through the integration of two theoretical perspectives related to the TAM and the JD-R models.” How was measured the impact of such technology on workers’ wellbeing and performance? Which indicators connected to wellbeing and performance were monitored or measured?  How was examined the role of job-related wellbeing in technology acceptance (lines 304-305)?

Thank you for the notes. Work engagement and work satisfaction represent positive work outcomes and are considered as wellbeing indicators in the literature (Schaufeli and Salanova, 2007), ). Furthermore, concerning the review related to performance, we underlined in the text that absenteeism can be considered a counterproductive behaviour that does not only impact the quality of work produced by the employee but also can negatively affect the activities of other employees in the organization and create undesirable risks for the employer. In the specific case of the transport sector, absenteeism has a very strong impact on customer service, which represents the main objective of economic effectiveness (pp. 2, line 66-68).

  1. The authors claim that “… work satisfaction increased…“ and „…which, in turn, prevented symptoms of strain and reduced the level of absenteeism.“  (lines 307-308). What is this claim based on? What symptoms of strain the authors of the paper investigated? Did they monitor changes in the satisfaction and absenteeism over time?

As you suggested, since we did not measure workers’ strain, we opted for greater caution and changed the sentence in terms of prevention of the potential symptoms related to technostressors.

  1. The authors state in the conclusion: “This study examined how different social factors influenced workers’ intention to use new technologyIn this regard, the first implication of this research was related to the role of perceived ease of use and usefulness, as a determinant of the willingness to use IT.” So, which social factors were examined?

Thank you for the suggestion.We specified that we examined the social support of colleagues, which represents a social factor of the organizational context.

  1. In conclusion, there are several statements that are not the result of the presented research, e.g. “Workers’ involvement, and their awareness of the impact of technology on performance effectiveness and its efficiency, fed positive expectations towards the use of new tools.” The most important findings of the presented research should be summarized in the conclusion.

We have moved the parts that referred to the most important findings of our study to the conclusion.

Round 2

Reviewer 2 Report

Congratulations to the authors. I am glad that their contribution entitled: The predictive factors of new technology adoption, workers ‘well-being and absenteeism: the case of a public maritime company in Venice has been fundamentally improved and adjusted. I have two suggestions for improvement:

  • a thorough formal correction according to jounal IJERPH instructions is needed,
  •  a thorough English correction is needed.

I wish the authors a lot of success. 

Author Response

Dear reviewer,

we would like to thank you for the quick reply. We corrected formal errors and did a linguistic review of the paper.

Best regards,

The Authors

Reviewer 3 Report

Thank you for sending a revised version of this manuscript.

I appreciate the effort of the authors to incorporate the comments and recommendations. I consider the current version to be much better elaborated and more sophisticated.

After reading the revised version, I have only a few small comments.

Formal editing of the paper requires review. Adjustments are needed e.g. in line spacing (lines 40-49), Hypothesis 6 is probably erroneously referred to as Hypothesis 3 (lines 211-212) etc.

I also recommend checking English phrases, for me it is e.g. incomprehensible “2.1.2. Studies Hypotheses”...

Also the following text (lines 249-250): “The chi-square test verified that there were no significant associations between all those variables and being a research participant sailor or not.” Since there is indicated on the lines 241-242: “This study adopted a convenience sample of 122 participants coming from a popula-241 tion of 175 sailors…“. Please explain or modify.

I wish the authors all the best for the next work.

Author Response

Dear reviewer,

we would like to thank you for the quick reply. We revised the paper as you suggested.

Best regards,

The Authors

REPLIES

  1. Formal editing of the paper requires review. Adjustments are needed e.g. in line spacing (lines 40-49), Hypothesis 6 is probably erroneously referred to as Hypothesis 3 (lines 211-212) etc.

As you requested, adjustments have been made and Hypothesis 6 has been corrected.

  1. I also recommend checking English phrases, for me it is e.g. incomprehensible “1.2. Studies Hypotheses”...

As you suggested we changed the phrase as follow: Aims and Hypotheses. An English check has been done.

  1. Also the following text (lines 249-250): “The chi-square test verified that there were no significant associations between all those variables and being a research participant sailor or not.” Since there is indicated on the lines 241-242: “This study adopted a convenience sample of 122 participants coming from a popula-241 tion of 175 sailors…“. Please explain or modify.

As you suggested, we changed the text: Moreover, we had compared the sailors that were research participants (70%) with sailors that didn't participate (30%), in the study, by the chi-square test. There were no significant associations between the two categories of sailors and gender, age, education, type of contract. Thus, the group of participants was substantially homogeneous concerning the other sailors who did not participate in the research for various reasons